# Inhibitory Control in Children with Tourette Syndrome Is Impaired in Everyday Life but Intact during a Stop Signal Task

**DOI:** 10.3390/jcm11020309

**Published:** 2022-01-08

**Authors:** Melanie Ritter, Signe Allerup Vangkilde, Katrine Maigaard, Anne Katrine Pagsberg, Kerstin Jessica Plessen, Julie Hagstrøm

**Affiliations:** 1Child and Adolescent Mental Health Center, Copenhagen University Hospital—Mental Health Services CPH, 2400 Copenhagen, Denmark; melanie.ritter@regionh.dk (M.R.); katrine.maigaard@regionh.dk (K.M.); anne.katrine.pagsberg@regionh.dk (A.K.P.); kerstin.plessen@chuv.ch (K.J.P.); 2Center for Visual Cognition, Department of Psychology, University of Copenhagen, 1353 Copenhagen, Denmark; signe.vangkilde@psy.ku.dk; 3Department of Clinical Medicine, Faculty of Health and Medical Sciences, University of Copenhagen, 2200 Copenhagen, Denmark; 4Division of Child and Adolescent Psychiatry, Department of Psychiatry, Lausanne University Hospital, University of Lausanne, 1011 Lausanne, Switzerland

**Keywords:** Tourette Syndrome, inhibitory control, ADHD, stop signal task

## Abstract

Tourette Syndrome (TS) has previously been associated with deficits in inhibitory control (IC). However, studies on IC in individuals with TS have produced conflicting results. In the present study, we investigated IC, comparing the Stop Signal Reaction Time (SSRT) measure with parent and teacher ratings of daily life IC in 169 children aged 8–12 (60 with TS, 60 typically developing controls, 27 with attention-deficit/hyperactivity disorder (ADHD), and 22 with TS + ADHD). We further investigated associations of IC with TS and ADHD symptom severity. Children with TS showed intact SSRT performance, but impairments in daily life IC, as reported by parents and teachers. For the latter, we observed a staircase distribution of groups, with the healthy controls presenting with the best IC, followed by TS, TS + ADHD, and finally ADHD. Dimensional analyses indicated a strong association between ADHD severity and both measures of IC. Our results indicate that children with TS are not impaired in a laboratory-based measure of IC, although some difficulties were evident from measures of everyday behaviour, which may in part be due to parents and teachers interpreting tics as disinhibited behaviour. Comorbid ADHD or the severity of subthreshold ADHD symptomatology appeared to account for IC deficits.

## 1. Introduction

Tourette Syndrome (TS) is a neurodevelopmental disorder, characterized by multiple motor tics and a minimum of one vocal tic, presenting before the age of 18 and lasting for at least one year [1,2]. Tics are defined as unwanted, sudden, rapid, and repetitive movements or sounds, which are often preceded by unpleasant physical sensations called premonitory urges.

Inhibitory control (IC) is an executive function defined as the ability to inhibit cognition and/or behaviour that is irrelevant or inappropriate to the execution of goal-oriented actions and adaptive behaviour [3,4]. IC is a broad construct encompassing (1) the cognitive inhibition of memories, thoughts, perceptions, and emotions, and (2) the behavioural inhibition of motor responses (response inhibition), compulsions, and immediate versus delayed reward [3]. IC is part of the impulsivity construct, which can be defined as the co-occurrence of impaired inhibitory processes and the experience of impulses, resulting in acts performed without delay, reflection, or voluntary direction [3].

TS has been theoretically conceptualised as a disorder of inhibition due to several clinical features of the disorder [5]. First, tics are characterized as semi-voluntary, since they can often be volitionally suppressed for at least a shorter period of time [6]. Second, the execution of unwanted and socially inappropriate acts, such as coprophenomena and non-obscene socially inappropriate behaviour, in spite of the ability to suppress them, could imply the disinhibition of motor actions [5]. Third, more than 90% of individuals with TS experience premonitory urges [7] and often describe tics as conscious actions to relieve the unpleasant sensation, which can be perceived as an inability to inhibit these interoceptive experiences [5]. Finally, TS has been linked to increased impulsive and disruptive behaviour [8], and approximately 50% of children with TS present with comorbid attention-deficit/hyperactivity disorder (ADHD) [7], which is characterised by impaired impulse and inhibitory control in children [9,10] and may additionally play a role in the disinhibition seen in TS. From a therapeutic perspective, cognitive and behavioural inhibition is central to in Habit Reversal Training (HRT) and Exposure and Response Prevention (ERP), which are the most effective behavioural treatment options for TS [11]. Both treatment types require the patients to inhibit their tics, which further emphasizes the need for investigating the extent of inhibitory difficulties in children with TS.

Empirical studies on IC in patients with TS have produced mixed results. A recent meta-analysis suggested mild but significant impairments in IC in individuals with TS compared to healthy controls in various performance-based inhibitory tasks, although subgroup analyses suggested an effect of comorbid ADHD [12]. Despite the association between TS and disinhibition, other studies have pointed to the opposite pattern, namely, an increased IC in TS. We previously found children with TS to be equal or superior regarding response inhibition to typically developing controls when inhibiting fast, impulsive actions [13], with another study demonstrating the increased activation of prefrontal areas in children with TS during tasks that require IC [14]. Together, these results support the hypothesis that children with TS develop neuroplastic compensatory mechanisms due to the ongoing suppression of tics [15,16].

Only a few previous studies on TS have included contrast groups of children with comorbid ADHD. One recent study indicated significantly impaired IC in the TS + ADHD group, whereas children with TS only did not differ from the healthy controls in a stop signal task [14]. Another recent study failed to demonstrate impairment in IC in children with TS + ADHD, as well as TS only, also in a stop signal task [17].

The aim of the present study was to investigate the specific association between TS and IC deficits assessed with an objective laboratory-based measure, as well as a more subjective parent- and teacher-reported questionnaire. The lab-based behavioural task thus measures response inhibition when stopping an ongoing motor response (action cancellation), reflecting latent level IC without the influence of impulsivity components. On the other hand, the rating-based questionnaire measures the ability to control impulses and stop inappropriate behaviour, thus reflecting the everyday presence of IC and related impulsive behaviour, as perceived by parents and teachers. To better understand the role of comorbid ADHD on inhibitory deficits in TS, we included children with TS, ADHD, and TS + ADHD, as well as typically developing controls. Furthermore, we explored associations between IC and the severity of tics, premonitory urges, and ADHD. We expected children with TS without ADHD to perform equally to typically developing controls, followed by children with TS + ADHD, and lastly ADHD. Finally, we expected IC to be associated with ADHD symptoms, but not with TS symptoms. A better understanding of the nature of (dis)inhibition in TS may have clinical implications related to the available therapeutic interventions, which largely depend on the ability of individuals with TS to withhold their tics.

## 2. Materials and Methods

The present study is an observational, cross-sectional, between-group, case-control design. 

### 2.1. Participants

The recruitment of participants took place from 2013 to 2016. Study participants were 169 medicine-naïve children aged 8 to 12 years (both inclusive). We previously reported on different data from the same cohort on topics of emotion regulation and response inhibition [13,18,19]. Children were assigned to two primary groups consisting of TS (*n* = 60) and typically developing children (*n* = 60), and two clinical contrast groups consisting of ADHD only (*n* = 27), and TS + ADHD (*n* = 22; Table 1). Exclusion criteria for all participants were birth at gestational age <37 weeks, the presence of any lifetime neurological condition, full-scale IQ below 80 (measured with The Wechsler Intelligence Scale for Children (WISC-IV) [20], and any current or prior use of psychotropic medication. For the clinical groups, further exclusion criteria were severe lifetime psychiatric comorbidity (autism spectrum disorder or psychotic disorders) and the presence of tics not qualifying for a TS diagnosis. For the control group, any lifetime psychiatric disorder was an exclusion criterion. Children in the clinical groups were recruited from a child and adolescent psychiatric outpatient clinic and a paediatric department in the Capital Region of Denmark. Children in the control group were randomly recruited via the Danish Civil Registration System [21] and matched for age and sex.

### 2.2. Clinical Measures

We screened all participants with the Kiddie-Schedule for Affective Disorders and Schizophrenia—Present and Lifetime Version (K-SADS-PL) [23], which is a semi-structured diagnostic interview assessing present and/or previous psychopathological symptoms, based on DSM-IV criteria [24]. In the healthy control group, we used the K-SADS-PL to confirm the absence of any lifetime psychiatric disorder, and in the clinical groups, it was used to confirm diagnoses of TS and ADHD.

Moreover, children who obtained a TS diagnosis from the K-SADS-PL underwent the Yale Global Tic Severity Scale (YGTSS) [25], which is a clinician-rated, semi-structured interview that scores the tic severity of motor and vocal tics over the past week, regarding number, frequency, intensity, complexity, and interference. The interview produces a total tic severity score and an impairment score, which together constitute a global tic score. The impairment score, however, has been criticized for being too subjective [26], and thus, results are reported with and without this score. Children with TS also completed a clinician-administered questionnaire, the Premonitory Urge for Tics Scale (PUTS) [27], which measures premonitory urges on ten items, assessing type, frequency, and control. For PUTS, in addition to the total score, we examined (1) any presence of premonitory urges, measured as a score of 3 or 4 in any of the first six questions representing different types of urges (yes/no) and (2) the ability to control tics, represented by a score of 3 or 4 in the final question (“I am able to stop my tics, even if only for a short period of time”; yes/no).

For all four groups, one parent and one teacher completed the ADHD Rating Scale (ADHD-RS), which measures ADHD symptoms and produces sub-scales for inattention and hyperactivity/impulsivity [28]. In the Danish version, an additional eight questions were added, covering symptoms of conduct disorder. Teacher ratings were primarily used in the diagnostic screening, whereas we included parent ratings in the analyses as T-scores based on Danish norms.

### 2.3. Measures of Inhibitory Control

#### 2.3.1. Stop Signal Task (CANTAB)

The stop signal task (SST) [4] is a popular, well-established tool for measuring response inhibition, in which the participants perform a primary two-choice reaction time (RT) task on each trial and are told to withhold their response when a stop signal is presented. The theoretical and mathematical horse-race model [4] conceptualizes the task as a race between the cognitive go- and stop-process, meaning that inhibition depends on which process finishes first. The primary measure, the stop signal reaction time (SSRT), which is the duration of the inhibition process, is estimated mathematically as it has no direct behavioural representation.

In the present study, we used hardware and software from The Cambridge Neuropsychological Test Automated Battery (CANTAB) [29], which offers an adaptive version of SST, applying the horse-race model and estimating SSRT. In this motor response version, participants sit in front of a CANTAB screen and press-pad with a left/right button. For the primary task, children were presented with a visual stimulus, consisting of a white arrow pointing either left or right, and are instructed to press congruently. They are instructed to withhold their response when cued by an auditory stop signal (a beep), which occurs shortly after the visual stimulus with a certain delay (stop signal delay; SSD). Subjects perform one practice block and five test blocks, each consisting of 64 trials, and the stop signal is presented in 25% of the trials (stop-trials). Since trials with no stop-signal (go-trials) occur more frequently, these constitute the imperative stimulus, making the corresponding go-response the dominating, prepotent response. On the other hand, the stop response is a so-called non-habitual response, requiring the subject to overrule a habituated response and cancel an already initiated motor-response. The software uses a tracking procedure to continually adjust SSD in stop-trials, adapting to the individual performances of the subjects. After successful inhibition, the SSD will increase (stop signal occurs later), making stopping more difficult, and after failed inhibition, SSD will decrease (stop signal occurs earlier), making stopping easier. As a result, the subject inhibits correctly on 50% of trials, fixing the race between the go- and stop-processes. The primary outcome is SSRT, which is calculated by subtracting the SSD by which 50% of stop-trials are correctly inhibited (SSD 50%) from the mean RT on go-trials (mean go-RT).

#### 2.3.2. Behaviour Rating Inventory of Executive Functioning (BRIEF)

RIEF is a parent- and teacher-rated questionnaire with 86 items, assessing children’s executive functions based on their everyday behaviour in two contexts (home and school) [30]. Each item describes a certain executive behaviour of the child, and the informant responds to the frequency of this behaviour on a three-point scale (“never”, “sometimes”, “often”). The summed scores are converted into T-scores, with higher scores indicating executive dysfunction. The BRIEF consists of eight clinical scales (Inhibit, Shift, Emotional Control, Initiate, Working Memory, Plan/Organize, Organization of Materials, and Monitor). In this study, we used the ‘Inhibit’ subscale (BRIEF-Inhibit) as a measure of participants’ IC. The BRIEF-Inhibit scale is based on 10 items (items 38, 41, 43, 44, 49, 54, 55, 56, 59, and 65), each measuring the child’s ability to control impulses and stop or regulate his/her own inappropriate behaviour. Some items describe acting more wildly and out-of-control than one’s peers in different settings, and others describe inappropriate or irrelevant verbal utterances. One parent and one teacher per participant filled out the questionnaire.

### 2.4. Data Analysis

Statistical analyses were performed with SPSS version 25.0, using two-tailed tests and an alpha level of 0.05. We tested all continuous outcomes for normality with the Shapiro–Wilk test and by visual inspection and used non-parametric tests when specific assumptions were violated. We controlled for the family-wise error rate with the Bonferroni correction where relevant. We aimed to achieve a power of 0.8 [31] for the SSRT analysis, which required a sample size of at least 45 participants in each of the primary groups, as calculated from effect sizes of around 0.30 from former studies [12] with the statistical software G*Power 3.1 [32].

We tested for differences in age and IQ with ANOVA and post hoc LSD tests and for differences in sex with the $\chi^{2}$ test. We assessed differences in tic onset and YGTSS scores with the *t* test, differences in ADHD-RS scores with the Kruskal–Wallis test and the Mann–Whitney test, and finally, we tested for differences in SES and comorbidity with Fisher’s exact test.

We removed values that differed by more than ± 2 SDs from the mean from the SSRT and BRIEF analysis. For group comparisons of SSRT, SSD, and BRIEF-Inhibit, we used the Kruskal–Wallis and the Mann–Whitney tests. Effect sizes (*r*) were calculated for the significant pairwise comparisons by dividing the test statistic (*z*) by the squared total sample size.

We assessed associations between inhibition (SSRT and BRIEF-Inhibit) and symptom severity and age of tic onset with Pearson’s product–moment correlation and the point–biserial correlation.

## 3. Results

### 3.1. Participant Characteristics

The four groups did not differ significantly in relation to age or sex, nor did the clinical groups differ regarding comorbidity, age of tic onset, or YGTSS (Table 1). The groups did, however, differ significantly with regard to IQ (*p* = 0.001) and SES (*p* < 0.001), with the highest scores in the control group, followed by TS, TS + ADHD, and finally the ADHD group. Furthermore, groups differed regarding ADHD-RS (*p* < 0.001), with most symptoms in the ADHD group, followed by TS + ADHD, TS, and finally the control group. Pairwise comparisons revealed that group differences in IQ and SES were driven by the ADHD group scoring significantly lower than the control group (*p* < 0.001 and *p* < 0.000, respectively) and the TS group (*p* = 0.001). Finally, group differences in ADHD-RS were driven by all groups differing in relation to inattention and hyperactivity (except for TS + ADHD and ADHD) and CD (except for TS versus controls and TS + ADHD versus ADHD).

### 3.2. Group Differences in Inhibitory Control

#### 3.2.1. Behavioural Data from the SST

One hundred sixty-three participants completed the SST, with three cases being excluded from the analysis due to the SSRT differing more than 2.5 SDs from the mean. SSRT was based on the last half of trials. Lower SSRT signifies a faster inhibition process, reflecting superior IC skills. SSRT and SSD were not normally distributed (*p* = 0.000; *p* = 0.002). It is noteworthy that the SSRT distribution was right-skewed, which is often the case for RT data and is in accordance with the horse-race model [4].

There was a significant main group effect on SSRT (H(3) = 17.339, *p* = 0.001; Table 2). Pairwise comparisons showed a significantly longer SSRT for ADHD, compared to the control group (*p* = 0.001, *r* = 0.4), and the TS group (*p* < 0.002, *r* = −0.4), with a medium to large effect. None of the other group differences were statistically significant, indicating that TS, TS + ADHD, and the healthy controls did not differ considerably regarding SSRT.

The proportion of successfully inhibited stop-trials was 0.53 on average, with no significant differences between groups, suggesting that participants in all four groups inhibited approximately 50% of the trials correctly, meaning that the race between the go- and stop-process was tied, which is a necessary assumption for SSRT to be estimated correctly (Table 2).

#### 3.2.2. Parent/Teacher-Reported Data from the BRIEF-Subscale

The BRIEF questionnaire was completed by 165 parents and 135 teachers. Two teacher-ratings were excluded from the analysis, since their composite BRIEF-Inhibit scores differed by more than 2.5 SDs from the mean.

There was a significant main group effect on BRIEF-Inhibit, as reported by parents (H(3) = 70.446, *p* < 0.001) and teachers (H(3) = 52.231, *p* < 0.001). For both, the control group had the lowest scores, followed by TS, TS + ADHD, and finally ADHD (Table 3). Pairwise comparisons of parental reports demonstrated significant differences between all groups, except for the two ADHD groups (TS < TS + ADHD, *p* < 0.001, *r* = −0.5; TS < ADHD, *p* < 0.001, *r* = −0.5; TS > control, *p* = 0.001, *r* = 0.3; TS + ADHD > control, *p* < 0.001, *r* = 0.7; ADHD > control, *p* < 0.001, *r* = 0.7). The same was the case for teacher reports (TS < TS + ADHD, *p* < 0.001, *r* = −0.4); (TS < ADHD, *p* < 0.001, *r* = −0.5); (TS > control, *p* = 0.005, *r* = 0.3); (TS + ADHD > control, *p* < 0.001, *r* = 0.7); ADHD > control, *p* < 0.001, *r* = 0.7).

### 3.3. Associations between Inhibitory Control and Symptom Severity

We did not find any significant associations between measures of IC and TS symptom severity (Table 4). Conversely, all correlations between IC and ADHD symptom severity were significant (*p* < 0.001) and remained significant with a Bonferroni-corrected threshold (0.05/9 = 0.006). The associations were positive, indicating that children with prolonged SSRT and increased BRIEF-Inhibit scores correspondingly displayed a high amount of ADHD symptoms. This result, however, appeared to be confounded by the presence of an ADHD diagnosis, as the associations between SSRT and inattention/hyperactivity were not significant when looking at the TS group only (inattention: *r* = 0.144, *p* = 0.280; hyperactivity: *r* = 0.051, *p* = 0.701). Results remained significant in the TS only group for the association between BRIEF and ADHD symptomatology (inattention: *r* = 0.510, *p* < 0.001; hyperactivity: *r* = 0.622, *p* < 0.001). Finally, the age of tic onset correlated inversely with SSRT; however, this did not remain significant when adjusting for multiple comparisons.

## 4. Discussion

The aim of the present study was to investigate the extent to which TS is associated with two aspects of IC deficits, namely, an objective, performance- and lab-based measure, and a daily life subjective parent- and teacher-reported questionnaire. For all group comparisons, we observed a staircase distribution of daily life IC, with the healthy control group presenting with the highest scores, followed by TS, TS + ADHD, and finally the ADHD group, although children with TS and the healthy controls did not differ significantly regarding the performance-based SSRT, in accordance with previous findings [14,17,33,34,35,36]. Conversely, children with ADHD showed a significant impairment in SSRT compared to healthy controls and children with TS. The parent- and teacher-reported BRIEF-Inhibit measure indicated a significant impairment in IC in children with TS compared to healthy controls, and the remaining groups differed significantly from each other, except for the TS + ADHD and ADHD groups. As expected, we found no association between IC and TS-related symptom severity (YGTSS and PUTS). However, we found significant associations between all three measures of IC- and ADHD-related symptoms (ADHD-RS), emphasizing that impaired inhibition was associated with symptoms of inattention and hyperactivity/impulsivity. Finally, the age of tic onset correlated with SSRT, suggesting that earlier tic onset may be associated with prolonged SSRT, although this result should be interpreted with caution, as it did not survive controlling for the familywise error rate.

The inconsistency between IC deficits in everyday life and in relation to the SST adds to the mixed results of previous studies. A possible explanation for this could be test-specific features representing different sub-domains of the complex construct of IC. Even within the domain of objective, lab-based measures, discrepancies are evident. In a recent study comprising largely the same group of participants as in the present study, we found IC to be superior in children with TS, although this was based on a different test of IC (a modified Simon task) and was carried out in an MRI scanner [13]. Overall, IC can be defined as the ability to inhibit thoughts and actions that are irrelevant to the execution of goal-oriented behaviour [4]. Besides consisting of multiple sub-domains, the concept is also strongly related to impulsivity [3], which overlaps with the BRIEF-Inhibit subscale that constitutes a broad measure of IC, as perceived by parents and teachers. The scale describes “deficient impulse control” in everyday life [30], and although these strongly related constructs can be discriminated neuropsychologically, it may not make sense to distinguish them in a natural setting. The SST is specific and objective, making it possible to draw conclusions about certain domains of IC. However, this specificity may compromise the ecological validity, since a laboratory setting is not representative of children’s everyday life, in which they are exposed to multiple contexts and triggers. The inclusion of the parent- and teacher-reports on daily life IC increases the ecological validity of the study but also subjects it to various biases associated with the use of questionnaires, such as recall bias and subjective thresholds for assessing problematic child behaviour. Finally, at least four of the 10 questions constituting the BRIEF-Inhibit scale cover aspects of behaviour resembling tics, such as having difficulties inhibiting one’s actions, which would understandably be difficult to distinguish for parents and teachers. Future studies should consider this duality of the questions, which may, in the present study, partly explain the significant impairment found in TS compared to the controls, which contrasts with the majority of previous studies on TS and IC. However, the present findings of parent- and teacher-rated daily life inhibitory deficits may still suggest that disinhibited behaviour could be part of the clinical presentation in children with TS and that this behaviour is consistent across different life contexts of family and school.

The present results consistently point to the notion that ADHD symptomatology in paediatric TS is associated with IC deficits. The staircase distribution of the groups on the BRIEF-Inhibit scale and the SSRT (however, not significant for the latter) indicates an objective impairment only when comorbid ADHD is present in addition to TS. Interestingly, children with TS + ADHD perform better than children with only ADHD, suggesting that TS somehow appears to be a protective factor with regard to IC deficits, and supporting the hypothesis that children with TS develop compensatory mechanisms to increase their IC. To further examine whether subthreshold symptoms of ADHD could explain IC deficits in the TS group, we repeated the correlation analysis in the TS group isolated from the remaining groups. Interestingly, the association between ADHD symptomatology and SSRT was small and non-significant when exploring only the TS group, while the correlation coefficient with the BRIEF-Inhibit scale was moderate and remained significant. This may indicate that for the objective and direct measure of IC, the presence of TS compensated for the presence of subthreshold ADHD symptoms. This was not the case for everyday behaviour, as perceived by parents and teachers, which could illustrate that the two measures represent distinct aspects of IC and supports the finding that the BRIEF-Inhibit scale is highly sensitive to ADHD [30].

The present SSRT results reflect an ability of children with TS to inhibit actions that are already initiated, possibly translating to the inhibition of tics by cancelling or compensating for the feeling of premonitory urges or the activation of motor behaviour. SSRT, however, reflects volitional inhibition, and thus the current study adds to the body of literature that indicates intact voluntary inhibition in TS, but does not take into account the potential deficits in automatic inhibition, which may play a part in tic generation; namely, tic generation and inhibitory deficits have been suggested to derive from an imbalance among volitional, goal-oriented actions, and automated, habitual action, with the latter appearing to be impaired in TS [37,38], although these constructs overlap. The ability to inhibit actions is essential to HRT, in which the child is trained to become more aware of the occurrence of a tic and to use competing responses to interrupt or inhibit the tic [39]. Moreover, a central feature of ERP is the child learning to weaken the association between premonitory urges and tics by exposing him/herself to the sensations and inhibiting responding with a tic, and thus the intact SSRT found in children with TS may reflect the necessary skills for participating in these first-line treatments. Of clinical relevance, this seemingly impaired ability in children with ADHD as well as TS with comorbid ADHD may hinder behavioural interventions targeted against tics and may prove a relevant target of evaluation before initiating these interventions. It would be interesting for future studies to investigate whether IC domains such as SSRT may serve as predictors of treatment outcome. None of the children in the present study had been treated with HRT or ERP, but the opposite would be interesting as well, namely, if participation in HRT/ERP increases IC.

The major strengths of this study were the inclusion of medicine-naïve children as well as clinical contrast groups of children with ADHD and TS + ADHD, in addition to the large primary groups of TS and typically developing controls. This design makes it possible to control for the effects of psychotropic medication and comorbid ADHD and focuses on the age group in which tics are most frequent. We recruited children in the control group randomly via the Danish Civil Registration System, thus increasing representativeness and reducing selection bias. Furthermore, the combination of performance- and rating-based measures, as well as dimensional measures of TS and ADHD symptomatology, enables a more nuanced investigation of IC across different contexts.

Several limitations exist as well. First, due to the modest sample sizes in the clinical contrast groups, the study may have been underpowered to detect between-group differences in these two groups. The small sample sizes were due to the focus on the two primary groups (TS and controls) and difficulty recruiting patients with ADHD from the psychiatric outpatient clinic. Second, the age of tic onset was not assessed systematically during the screening, but collected retrospectively from the K-SADS summary, and the variable was calculated for only a limited portion of the sample (72% of the TS group and 73% of the TS + ADHD group). Additionally, we did not screen systematically for the genetic or immunological causes of tics, but all patients underwent a physical screening. Third, the use of specific samples while controlling for comorbidity could be a limitation as well as a strength, since these samples may not be as representative of the population based on which we aim to draw conclusions. The inclusion of dimensional measures of TS and ADHD symptom severity, however, nuances the findings and allows for analyses of subthreshold symptoms of psychopathology. The ADHD group presented with significantly lower IQ than the control group, which, however, reflects the general presentation of the disorder. We thus chose not to include IQ as a covariate in the analysis, since this would compromise the representativity of the ADHD sample [40]. Furthermore, although the inclusion criterion of IQ > 80 has been applied in other studies in the field of TS, this cut-off is not theoretically founded and may reduce the representativeness of the sample. Only four patients in total, however, were excluded due to IQ < 80. Finally, OCD has been associated with impaired SSRT [41], which may influence the results. In the present study, however, only six patients in total presented with comorbid OCD (three in the TS group and three in the TS + ADHD group), and thus we did not control for this in our analyses.

In conclusion, our mixed report results support the hypothesis of intact response inhibition in children with TS and suggest that these children are able to volitionally inhibit motor actions when they act in a goal-directed manner. Simultaneously, children with TS presented with inhibitory deficits in their everyday life, which seemed to relate strongly to ADHD symptomatology and may in part be explained by the difficulty distinguishing tics from other types of disinhibited behaviour. Future studies could advantageously examine this discrepancy further, as well as the subtypes of the broad concept of IC and their relation to treatment outcomes to not only increase the understanding of potential mediators of treatment effectiveness, but also enable a realistic assessment of treatment eligibility before embarking on the highly demanding treatment options of HRT and ERP.

## Figures and Tables

**Table 1 jcm-11-00309-t001:** Participant characteristics for the total sample.

Characteristic	Controls	TS	TS + ADHD	ADHD	*p*
	(*n* = 60)	(*n* = 60)	(*n* = 22)	(*n* = 27)	
Mean age, years	9.90 (1.3)	9.97 (1.3)	9.50 (1.3)	9.60 (1.1)	0.366
Male, *n* (%)	48 (80.0)	49 (81.6)	17 (77.3)	19 (70.4)	0.678
Mean IQ	103.9 (11.2)	102.6 (9.9)	98.8 (10.9)	94.4 (8.5)	0.001
Mean SES	5.98 (1.4)	5.72 (1.5)	4.95 (1.8)	4.25 (1.8)	<0.001
Comorbidity, *n*					
CD (%)	0 (0)	0 (0)	1 (4.5)	1 (3.7)	0.200
ODD (%)	0 (0)	4 (6.6)	5 (22.7)	6 (22.2)	0.050
OCD (%)	0 (0)	3 (5.0)	3 (13.6)	0 (0)	0.136
GAD (%)	0 (0)	2 (3.3)	0 (0)	0 (0)	1
SAD (%)	0 (0)	2 (3.3)	0 (0)	1 (3.7)	1
Phobia (%)	0 (0)	4 (6.6)	0 (0)	0 (0)	0.398
Tics NOS (%)	0 (0)	0 (0)	0 (0)	1 (3.7)	0.450
TS age of onset	NA	5.9	6.2	NA	0.575
YGTSS					
Global	NA	36.5 (11.1)	37.3 (18.5)	NA	0.813
Total	NA	19.7 (6.3)	20.3 (7.7)	NA	0.707
ADHD-RS					
Inattention	49 (12)	57 (13)	78 (15)	86 (15)	<0.001
Hyperactivity/impulsivity	45 (10)	54 (11)	77 (17)	82 (18)	<0.001
CD	48 (9)	54 (14)	71 (22)	71 (21)	<0.001

The table includes the entire sample participating in either SST or BRIEF. TS = Tourette Syndrome. ADHD = attention-deficit/hyperactivity disorder. SES = socioeconomic status of participants’ parents classified using the International Standard Classification of Education (ISCED) [22]. CD = conduct disorder. ODD = oppositional defiant disorder. OCD = obsessive-compulsive disorder. GAD = generalized anxiety disorder. SAD = separation anxiety disorder. Tics NOS = tics, not otherwise specified.

**Table 2 jcm-11-00309-t002:** Group differences in the stop signal task.

	Controls	TS	TS + ADHD	ADHD	Total	*p*
	(*n* = 56)	(*n* = 58)	(*n* = 21)	(*n* = 25)	(*n* = 160)	
RT						
SSRT	209 (61)	213 (62)	227 (54)	269 (66)	222 (64)	0.001
SSD (50%)	348 (155)	325 (146)	344 (116)	323 (151)	335 (146)	0.837
Errors						
Suc. stop	0.53 (0.1)	0.51 (0.1)	0.55 (0.1)	0.53 (0.1)	0.53 (0.1)	0.403
Com. errors	5.6 (8.2)	6.0 (6.5)	8.7 (7.4)	10.7 (14.7)	7.0 (9.0)	0.054

TS = Tourette Syndrome. ADHD = attention-deficit/hyperactivity disorder. SSRT = stop signal reaction time in *ms*. Suc. stop = the proportion of successfully inhibited stop trials. SSD (50%) = stop signal delay, when half of the trials are correctly inhibited. Com. error = committed errors in the reported direction of the arrow.

**Table 3 jcm-11-00309-t003:** Group differences in the BRIEF-Inhibit scale.

	Controls	TS	TS + ADHD	ADHD	Total	*p*
Parent (*n*)	(60)	(60)	(22)	(23)	(165)	
Teacher (*n*)	(44)	(52)	(19)	(18)	(133)	
Parent						
Inhibit	44.9 (6.2)	50.4 (8.3)	64.2 (10.6)	67.4 (13.1)	52.6 (12.2)	<0.001
Teacher						
Inhibit	48.2 (5.9)	54.8 (11.1)	67.0 (10.0)	67.8 (8.9)	56.1 (11.8)	<0.001

TS = Tourette Syndrome. ADHD = attention-deficit/hyperactivity disorder. Inhibit = BRIEF subscale for inhibition (lower scores indicate better inhibitory control).

**Table 4 jcm-11-00309-t004:** Correlations between inhibitory control and symptom severity.

	SSRT		BRIEF—Parent		BRIEF—Teacher	
	*r*	*p*	*r*	*p*	*r*	*p*
YGTSS—Global	−0.052	0.655	−0.024	0.836	−0.141	0.243
YGTSS—Total	−0.004	0.972	−0.116	0.306	−0.147	0.150
PUTS—Total	−0.214	0.059	−0.303	0.006	−0.070	0.563
PUTS—Presence of any	−0.107	0.350	−0.097	0.388	−0.146	0.229
PUTS—Hold back	−0.130	0.258	−0.142	0.207	0.055	0.652
ADHD-RS—Inattention	0.330	<0.001	0.733	<0.001	0.416	<0.001
ADHD-RS—Hyperactivity/imp.	0.303	<0.001	0.833	<0.001	0.464	<0.001
ADHD-RS—CD	0.297	<0.001	0.694	<0.001	0.365	<0.001
Tics—age of onset	−0.295	0.027	−0.213	0.105	0.237	0.087

The table includes the entire sample participating in either SST or BRIEF. Table presents unadjusted *p* values. SSRT = stop signal reaction time in *ms*. BRIEF = Behaviour Rating Inventory of Executive Function, Inhibit subscale. YGTSS = Yale Global Tic Severity Scale. PUTS = Premonitory Urge for Tics Scale. ADHD-RS = ADHD Rating Scale. CD = conduct disorder.

## Data Availability

The data presented in the present study is available upon request to all researchers.

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
