# Peer review of "Inhibitory Control in Children with Tourette Syndrome Is Impaired in Everyday Life but Intact during a Stop Signal Task"

_jcm, 2022, doi:10.3390/jcm11020309_

Round 1

Reviewer 1 Report

The authors investigate inhibitory control in a large sample of children with tic disorders.

Impressively, there are a number of appropriate control groups, which account for ADHD. They find that voluntary reactive inhibition is not impaired in patients with Tourette syndrome but is by the presence of ADHD. I like how the authors have addressed the ecological validity of their study by simultaneously recording real-life manifestations of impaired behavioural inhibition. I would be happy for the manuscript to be published with only minor comments, as outlined above:

  1. It is becoming clearer that voluntary aspects of movement control (preparation, execution and inhibition) are normal in patients who have tics. This current study adds to that growing body of literature. However, there is evidence that tics may be caused by abnormal automatic inhibition, which lies outside voluntary control (see https://doi.org/10.1093/brain/awaa024). I suggest that the authors discuss alternative mechanisms of tic generation such as impairments in automatic inhibition.
  2. The authors correlate several behavioural and clinical measures (Table 4), Please state if the comparisons were corrected for multiple comparisons and if so, how many.
  3. OCD can also influence SSRT (see https://pubmed.ncbi.nlm.nih.gov/27334752/) but is not adjusted for in these comparisons. The authors may wish to address this in the discussion or correct for them.
  4. The effects on inhibitiory control may be an effect of the task used. Indeed, the same group have published that inhibitory function as assessed with the Simon  task shows  superior inhibitiory performance (see https://www.sciencedirect.com/science/article/pii/S0028393219301204?via%3Dihub). The authors may wish  to address this caveat.

Reviewer 2 Report

The authors present a very interesting study approaching the subject of Executive Functions (and in particolar of Inhibitory Control) in subjects with Tourette syndrome. The idea to evaluate the role of ADHD in this context is a very good one, given the high comorbidity between Tourette syndrome and ADHD and the fact that both disorders have been related to problems with Executive Functions.

The manuscript is globally well written, and I especially liked the clear power calculation.

Still, I have a few points to raise.

Major points

In the Introduction nothing is saida about the so-called NOSI behaviors (Non-Obscene Socially Inappropriate). They are however conceptualized as failure of inhibition, just as much as coprophenomena.

In the Methods, I'm hapy to read well described inclusion and exclusion criteria. However I wonder 1) why were data collected more than 5 years ago and never published since then and 2) why was a full scale IQ>80 choosen (I could have understood >70 or even >85 easily...).

Was the possibility of tic-causing disorders (e.g. Wilson syndrome or PANS) explored?

The authors calculate that 45 subjects per group were needed. Still, two groups are clearly smaller. I aknowledge that the authors declared this as a limitation of the study, but I think that a bit more of explanation on why this happened is neeeded.

How did the authors managed the frequent psychopathological comorbidities present in Tourette syndrome?

Minor points

Page 2: "disorder of disinhibition": I think the authors meant "disorder of inhibition"

The authors name HRT as a gold standard treatment. There are however some data not in line with this vision, e.g. obtained in Italian children. Could the author discuss this?
